# Chronic Community Exposure to Environmental Metal Mixtures Is Associated with Selected Cytokines in the Navajo Birth Cohort Study (NBCS)

**DOI:** 10.3390/ijerph192214939

**Published:** 2022-11-13

**Authors:** Nicole Thompson González, Jennifer Ong, Li Luo, Debra MacKenzie

**Affiliations:** 1Integrative Anthropological Sciences, University of California Santa Barbara, Santa Barbara, CA 93106, USA; 2Department of Anthropology, University of New Mexico, Albuquerque, NM 87131, USA; 3Academic Science Education and Research Training Program, Health Sciences Center, University of New Mexico, Albuquerque, NM 87131, USA; 4Health Sciences Center, College of Pharmacy, University of New Mexico, Albuquerque, NM 87131, USA; 5Department of Mathematics and Statistics, University of New Mexico, Albuquerque, NM 87131, USA

**Keywords:** immunomodulation, inflammation, immunosuppression, pollutants, mining waste, tribal lands

## Abstract

Many tribal populations are characterized by health disparities, including higher rates of infection, metabolic syndrome, and cancer—all of which are mediated by the immune system. Members of the Navajo Nation have suffered chronic low-level exposure to metal mixtures from uranium mine wastes for decades. We suspect that such metal and metalloid exposures lead to adverse health effects via their modulation of immune system function. We examined the relationships between nine key metal and metalloid exposures (in blood and urine) with 11 circulating biomarkers (cytokines and CRP in serum) in 231 pregnant Navajo women participating in the Navajo Birth Cohort Study. Biomonitored levels of uranium and arsenic species were considerably higher in participants than NHANES averages. Each biomarker was associated with a unique set of exposures, and arsenic species were generally immunosuppressive (decreased cellular and humoral stimulating cytokines). Overall, our results suggest that environmental metal and metalloid exposures modulate immune status in pregnant Navajo women, which may impact long-term health outcomes in mothers and their children.

## 1. Introduction

The legacy of mining from the Cold War Era has left several U.S. communities exposed to metal and metalloid contaminants (hereafter “metals”) with potential health consequences. The abandonment of more than 160,000 hard rock mines and >500,000 mine waste sites (Figure 1) [1] has resulted in 40% of watersheds in the western United States being contaminated with mine waste and related metals [2]. These sites are often located on or contiguous with the watersheds of tribal lands, such as the Navajo Nation. Although Navajo communities have long been concerned that environmental exposure to mine waste contributes to poor health outcomes among tribe members, no comprehensive characterization of the metal body burden of this population has been conducted prior to the Navajo Birth Cohort Study (NBCS). Here, to address potential immune modulation resulting from environmental exposures, we characterize the relationship between measured metal and cytokines concentrations among pregnant Navajo women participating in the Navajo Birth Cohort Study (NBCS).

Members of the Navajo Nation, located in the Four Corners Region of the Southwestern US (Figure 1), are exposed to the metal contaminants resulting from historic mining activities via several pathways. While active mining and milling on Navajo Nation for uranium ended in 1986, more than 500 abandoned mines and waste sites remain, containing multiple metals and metalloids, including arsenic (As), lead (Pb), cadmium (Cd), manganese (Mn), and sometimes mercury (Hg) [3,4,5]. Navajo community members are likely exposed to such wastes via multiple pathways, including the consumption of local water and crops [6], contact with contaminated soil and dust from mine features, and the inhalation of metals released from combustion for home heating and cooking and local power sources [7,8]. Exposure via drinking water is of particular concern, as >30% of Navajo members lack access to public water systems, and unregulated water sources exceed the maximum contaminant levels of the Safe Drinking Water Act for uranium by 13% and arsenic by 15% [9]. Even among major public water systems on Navajo Nation, many have been repeatedly out of compliance according to standards for one or more metals [10].

Metal exposures are well linked to several adverse health effects. Chronic low-dose exposures to uranium and arsenic in drinking water, for example, are associated with cancers [11] and various aspects of metabolic syndrome and autoimmunity [12,13], including chronic inflammation [14], cardiovascular diseases [15,16], and kidney damage [17]. Co-occurring metals such as lead, cadmium, manganese, and mercury are related to a similarly wide array of negative health outcomes, further including hypertension and neurodegenerative disease [7,18,19,20,21]. Prenatal exposures to metals are also related to elevated risks of infant mortality [22,23], stunted growth [24], and lung cancer in adulthood [25]. The health effects of metal exposure are, therefore, not localized to a single generation.

A primary mechanism by which metals influence the development of disease risk is likely their immunomodulatory effects [26,27]. Several metals are immunostimulatory [18,19,28,29] and others immunosuppressive [25,30], with effects often varying by dose, the chronicity of exposure, and individual factors, such as age and sex [18,31,32]. Uranium and arsenic exposures are linked to alterations in cellular and humoral immune responses in animals [33,34,35] and in humans with occupational metal exposure [36,37]. Less is understood, however, about the role of these metals in immune system alterations that occur in populations chronically exposed to environmental metals [38,39,40]. Studies in such systems are important to reveal the community-wide effects of metal exposures, the epidemiology of associated diseases, and steps for policy action. In addition, such studies in pregnant women lend insight into the maternal–fetal immune environment, a key factor in the developmental origins of health and disease [41,42].

In this study, we examine the potential immunomodulatory effects of specific environmental metals and metalloids among pregnant women participating in the Navajo Birth Cohort Study. Decades of research firmly establish that members of the Navajo Nation experience higher rates of severe respiratory infection [43], end-stage renal disease [44], metabolic syndrome [45,46,47], and infant mortality [48] relative to U.S. averages. Given Navajo members’ chronic low-level exposure to environmental metal mixtures [9,49], we expect that metal exposures play a role in observed health disparities. Specifically, we hypothesize that metal exposures exert immunomodulatory effects in exposed individuals. To test this, we pair participant biomonitoring (U, As and its species AsIII, MMA, and DMA; Hg, Pb, Cd, and Mn), and circulating cytokine and acute phase protein levels (IL-4, IL-6, IL-7, IL-10, IL-12, IL-17, IL-29, IFNα, IFNγ, TNFα, and CRP), to test the relationships of metal and metalloid exposures with the key mediators of innate, cellular, and humoral immune function.

## 2. Methods

### 2.1. Inclusion Criteria

The NBCS was initiated to address Navajo community members’ concerns about the potential health effects of chronic environmental exposure to uranium mine wastes. In 2013, the NBCS began recruiting pregnant women between 14 and 45 years of age who had lived on Navajo Nation for at least 5 years. To be included in the study, women had to be willing to deliver at a participating hospital, and have their child followed up for one year postnatally.

### 2.2. Survey Information

At enrollment, a survey of socioeconomic, demographic, lifestyle, and reproductive history information was administered by trained community health environmental research staff. The responses were entered into the research database RedCap in accordance with all privacy and security requirements of both the Navajo Nation (NNHRRB 11-323) and University of New Mexico institutional review boards (UNM HRPO 11-310). Women broadly represented several demographic categories (Table 1) and were recruited during all trimesters of pregnancy.

### 2.3. Biological Sample Collection

Trained hospital staff collected and prepared blood and urine samples. Samples intended for the metal biomonitoring analysis were collected using pre-screened metal-free cups, transfer pipettes, and Nalgene cryo-vials provided by the Center for Disease Control (CDC), National Center for Environmental Health (NCEH), and the Division of Laboratory Sciences (DLS). At enrollment or 36 weeks gestation prenatal visits, the participants provided 40–50 mL of urine in a sterile collection cup. Laboratory staff aliquoted 1.8 mL urine samples into separate Nalgene cryo-vials for multi-element metal, total arsenic, speciated As, creatinine, and mercury/iodine analysis. Hospital laboratory staff also collected peripheral blood via venipuncture and then allowed the blood to clot at room temperature for 30 to 40 min. Once clotted, laboratory staff centrifuged the blood tube at 2400 revolutions per minute for 15 min to separate the serum. Aliquots of serum (1.8 mL) were transferred to a 2.0 mL Nalgene cryo-vial. After processing the urine and serum samples, hospital staff placed all cryo-vials in a −80 °C freezer for storage. The samples were transferred from participating facilities on Navajo Nation on dry ice to freezer storage facilities at UNM. Chain of Custody forms were completed, reviewed, and validated at each stage of collection, storage, and analysis. The samples were stored, analyzed, and disposed of in accordance with participant wishes, as indicated on the consent forms. Samples not consumed during analysis will be returned to participants after the close of the study analysis period if they desire, in accordance with cultural practices.

### 2.4. Metal Biomonitoring

UNM laboratory staff examined the samples for quality control. Then, the urine vials and one serum vial were shipped on dry ice to the CDC Division of Laboratory Systems for analysis. Urine concentrations of metals and metalloids (U, As and its species Arsenite AsIII, Monomethylarsonic acid MMA, Dimethylarsinic acid DMA, Pb, Cd, and Mn) were measured using inductively coupled plasma dynamic-reaction cell mass spectrometry [ICP-DRC-MS Perkin Elmer NexION 300, CDC Method No. 3018.6-02, 3018A4-02, & 3031.1-01; [50,51,52,53]. HPLC was used for arsenic speciation prior to the ICP-MS analysis (CDC Method No. ITU003B). HPLC used an Agilent 1200 Series thermostatted column compartment (Item #G1316B) along with an Agilent Instant Pilot control module (Item #G4208-67001). Serum concentrations of the metals (Pb, Cd, Mn, Hg) were also measured using ICP-DRC-MS [CDC Method No. 3016.8-05] [54]. Metal exposures for Pb, Cd, and Mn were examined in both serum and urine, as their presence in the different media is known to vary by the dose and chronicity of exposure [55,56,57]. To determine if the ICP-MS runs were in control, the multi-rule quality control system (MRQCS) was implemented [58] based on the results of two to three custom-made and characterized urine bench quality control materials that were inserted at the start and end of each analytical run. All reference materials were made in-house at the CDC, according to NIST standards. The limit of detection for these elements in urine and serum ranged from 0.002 to 1.91 ng/mL, depending on the analyte and biological media (complete LoD by element in Table 2). The CDC’s acceptable percent of recovery ranges from 80 to 120%. Urinary metal concentrations were adjusted for sample creatinine concentrations, determined using a Roche/Hitachi Modular P Chemistry Analyzer.

### 2.5. Serum Cytokines

Eleven biomarkers, including 10 cytokines and 1 acute phase protein, were analyzed in the serum samples: interleukin-4 (IL-4); interleukin-6 (IL-6); interleukin-7 (IL-7); interleukin-10 (IL-10); interleukin-12p70 (IL-12); interleukin-17 (IL-17); and interleukin-29 (IL-29); interferon alpha (IFNα); interferon gamma (IFNγ); tumor necrosis factor alpha (TNFα); and C-reactive protein (CRP). These 11 markers were chosen for early analysis because they are indicative of immune responses from innate, cellular, and humoral branches, and their irregularities are associated with conditions of chronic inflammation and related cardiovascular and metabolic disease [59,60,61,62].

Cytokine measurements were performed using the Meso Scale Discovery multiplex four- or ten-spot 96-well electrochemiluminescence detection platform. Serum samples were diluted 1:2 in appropriate assay buffer and incubated on the plate for two hours. The plates were washed with PBS-Tween, and corresponding Sulfo-Tag secondary reagents were added. They were then incubated for an additional two hours. Plates were read using the MESO QuickPlex SQ 120 microplate reader and results calculated using the Discovery Workbench 4.0 software. Urine and serum metal measurements below the limit of detection (LoD) were replaced by the (LoD) divided by the square root of two.

### 2.6. Statistical Analysis

To broadly describe their associations, we evaluated correlations among the metals and between metal concentrations and cytokine levels (alpha = 0.05). To control for confounding variables in metal–cytokine relationships, we then used multivariate linear regression models. Metal biomonitoring and cytokine concentrations were log-transformed to reduce skewness. Certain cytokines varied by participant age and trimester in univariate regression; thus, we included these variables as analytical controls in all multivariate regressions (unpublished data). We further controlled for pre-pregnancy BMI class in multivariate models of CRP, as the two were strongly positively related. Cytokine values did not differ by education level, marital status, annual income, or fetal sex in univariate models, apart from weak relationships with IL-10, which were best accounted for by age; thus, these variables were not controlled for in multivariable modeling. We performed backwards selection from full models (including participant attributes and all log-transformed metals) based on AIC criteria with the R function “step”.

## 3. Results

### 3.1. Participant Metal Concentrations and Their Correlations

Four metal concentrations were on average higher among participants in this sample than in the NHANES participants: uranium, arsenic species (i.e., AsIII, MMA, DMA), cadmium, and manganese (Table 2). Concentrations that fell below the NHANES medians were those of lead and mercury. Metals demonstrated several non-transitive correlations among one another (Figure 2, Appendix A). Urinary or blood lead concentrations were positively correlated with all metals except urinary cadmium, urinary magnesium, and mercury. Uranium was strongly and positively correlated with arsenic and its species, lead, and weakly with urinary magnesium. Arsenic and its species were also positively correlated with urinary manganese. Apart from its relationship with uranium, arsenic, and lead, manganese was positively correlated with cadmium. Mercury was unrelated to any metal, perhaps because of its overall low presence among the participants.

### 3.2. Participant Cytokine Levels and Correlations with Metal Exposures

The average cytokine levels among the study participants are summarized in Appendix A. Concentrations of arsenic and its species, lead, cadmium, and manganese were correlated with several cytokines among the participants (Figure 3, Table 3). Most notably, arsenic species were negatively correlated with Th2 cytokine IL-4, inflammatory cytokines IL-6, IL-12, IL-29, and TNFα, and complemented CRP. The arsenic species DMA, however, was positively correlated with IFNα.

### 3.3. Multivariable Linear Regression Models

Due to the exploratory nature of this analysis, we did not correct for multiple testing and consider all the metal exposures in the best fit models to have notable relationships with cytokine levels, although we emphasize those with particularly strong relationships (*p* of ß < 0.05, Table 4). This helps target potentially important exposures for future analysis. All cytokines demonstrated relationships with metal exposures (Table 4). Although we have structured Table 4 according to model structure, where cytokine levels are responses, we have organized the presentation of the results and their discussion according to each metal and its profile of relationships with various cytokines.

Arsenic species, to which the participants had high exposures, demonstrated broadly immunosuppressive effects on cytokines (Table 4). Higher levels of total arsenic, AsIII, and MMA together corresponded with lower IL-4 and IL-29. Arsenic’s relationship with CRP depended on the species considered, as MMA and total arsenic had a strong negative association with CRP, whereas DMA had a strongly positive association. DMA further had a strong, positive relationship with IFNα. Counter to expectations, uranium demonstrated only a weakly negative relationship with IFNγ, despite the participants’ high exposure.

For certain metals, blood and urinary measures demonstrated opposite immunomodulatory effects on the same cytokine (Table 4). Blood lead, for example, was broadly inflammatory with strongly positive relationships with IL-10, IL-17, IFNγ, and TNFα; however, urinary lead decreased TNFα. Blood cadmium, further, appeared immunosuppressive in its negative relationship with IL-4, IL-29, and TNFα; however, urinary cadmium increased both IL-29 and TNFα. Other effects of the metals were more straightforward. Manganese demonstrated a general immunostimulatory influence, including strong positive relationships with IL-4, IL-12, and TNFα. Lastly, mercury exposures did not have any strong relationships with cytokines, apart from a weakly positive relationship with IFNγ.

## 4. Discussion

The aim of our study was to probe the immunomodulatory effects of long-term metal exposure among pregnant women participating in the Navajo Birth Cohort Study. This is particularly important as the abundance of environmental uranium and arsenic in the Navajo Nation has been a health concern among its members for decades, and because the prenatal immunological environment is a critical pathway for the developmental origins of health and disease. Here, we find that exposures to metal mine wastes demonstrate several strong relationships with inflammatory and autoimmune-related cytokines among pregnant women. In particular, arsenic appeared to be generally immunosuppressive, lowering IL-4 and IL-29, but with variable effects on inflammatory cytokines IFNα and CRP depending on its species. Overall, our results build on findings from Erdei et al. [12] and Ong et al. [63] to recommend future avenues for targeted examination of the effects of combination, dose, and chronicity of environmental metal exposures on immune function. Here, we discuss the consistency of our findings with the existing literature, their implications for health outcomes among current and future members of the Navajo Nation, and future directions of this work.

### 4.1. Immunomodulatory Effects of Metals and Comparisons with Previous Literature

The cytokine profile associated with arsenic exposure was nuanced, such that arsenic and early-state species had a general immunosuppressive effect (lower IL-4, IL-29, and CRP), but its end-state species appeared to increase certain inflammatory cytokines (DMA, higher IFNα and CRP). The immunosuppressive effects of arsenic and its species are well documented [25,34]: even at low doses, AsIII in particular corresponds with a greater disruption of T cell proliferation and inhibition of DNA repair [64], and its metabolite MMA plays a key role in the disruption of IL-7 signaling in mice [65]. Our findings of arsenic-related immunosuppression also align with evidence in Navajo women that arsenic exposure reduced the prevalence of autoantibodies [12]. Participant sex may further shape such findings. Women have greater methylation capacity of arsenic than men, which further accelerates during pregnancy, leading women to suffer the immunotoxicity of arsenic less severely [31,66]. The immunosuppressive effects of arsenic in our study are, thus, potentially reduced given our focus on women during pregnancy.

Although arsenic is known to suppress adaptive immunity, in some cases it can increase innate inflammatory responses. In mice, for example, chronic arsenic exposure reduced mature macrophage numbers and increased immature and pro-inflammatory macrophages [35]. Further, in humans, arsenic increased viral responses [26], inhibited DNA repair [67], and chronic low levels increased the expression of pro-inflammatory mediators [14]. This type of pro-inflammatory shift may lie at the root of our findings of positive relationships of DMA with IFNα.

Other relationships between biomonitored metals and cytokines in this study are consistent with evidence in previous literature. Lead’s strong positive relationship with inflammatory markers among the participants in this study (IL-10, IL-17, IFNγ, TNFα) is consistent with its particularly strong pro-inflammatory effects in women [18,68] and is suggestive of participants’ chronic exposure [69]. Concentrations of cadmium among the participants corresponded with lower IL-4, suggesting an immunological shift from a Th2 towards a Th1 response. Opposite directions of influence of blood and urinary cadmium on IL-29 and TNFα perhaps results from differences in chronicity that measures capture, where blood represents acute and urine long-term exposure [56]. These effects could also represent differences in dose in chronic vs acute exposures, as low cadmium can lead to proinflammatory effects and high cadmium exposure the opposite [70]. Finally, manganese appeared to be a general immunostimulant in our study (higher IL-4, IL-12, TNFα), consistent with the effects of chronic manganese exposure on increased inflammatory, humoral, and anti-tumor activity in murine models [27] and increased IL-1ß in pregnant women [71].

### 4.2. Implications for Health Outcomes

Our results demonstrate clear associations between environmental metal exposures and immune activity, suggesting that metals’ immunomodulatory effects combined with chronic and potentially life-long exposures contribute to health outcomes in current and future Navajo community members. Uranium and arsenic are primary concerns, given community members’ high exposures. Although uranium did not appear to modulate immunity, the broad immunosuppression of arsenic could pave the way for infectious disease and overreliance on pro-inflammatory immune responses. In combination with the immunostimulatory effects of lead and manganese (Figure 2), immunomodulation by metals likely contributes to the prevalence of disease in the Navajo Nation via distinct and complementary pathways [35]. Our study also highlights concerns of the intergenerational effects of maternal metal exposures on offspring. Arsenic and its species readily cross the placenta, increasing infant mortality [22,23] and later risks of lung cancer [25]. Arsenic, lead, and cadmium exposures during pregnancy all result in fetal immunosuppression and lower thymic volume [26]. In turn, prenatal exposures to uranium reduce offspring growth [24]. The potential reach of maternal metal exposures, therefore, is far and the potential consequences dire.

## 5. Limitations and Future Directions

We acknowledge that the sample size of this study was limited, including that biological samples were collected at a single time point. In addition, studies of environmental metal exposures are scarce, relative to studies of experimental and occupational exposures; therefore, definitive comparisons between our findings and previous work are few. Nevertheless, significant changes in circulating cytokines with metal biomonitoring concentrations strengthen the case that environmental metal exposure contributes to immune system disruption. Our findings encourage several additional approaches. A first approach is to determine whether findings are broadly generalizable to women by longitudinally analyzing a larger sample of women. This would include examining potential alterations in arsenic metabolism (e.g., AsIII to MMA to DMA) with chronicity of exposure. As individuals are exposed to distinct sets of metals in their environment, a second approach is to examine metals’ combined influences on immune function. This would include examining how potentially beneficial metal exposures, such as zinc [72], ameliorate the damaging effects of other metals. Finally, the findings urge implementing plans to reduce the risks that environmental metal exposures pose to personal and larger scale environmental health.

## 6. Conclusions

Using data from the Navajo Birth Cohort Study (NBCS), the aim of this study was to evaluate the immunomodulatory effects of environmental metal exposures among Navajo Nation community members. Highly correlated arsenic species in biomonitoring appear to be strong drivers of immunosuppression. Taken as a whole, our results suggest that chronic community-level exposure to mixed metals and metalloids on the Navajo Nation plays a role in altering immune responses. The findings of our study are relevant to both acute immunomodulation and its potential long-term and widespread consequences, informing both basic science and health policy.

## Figures and Tables

**Figure 1 ijerph-19-14939-f001:**
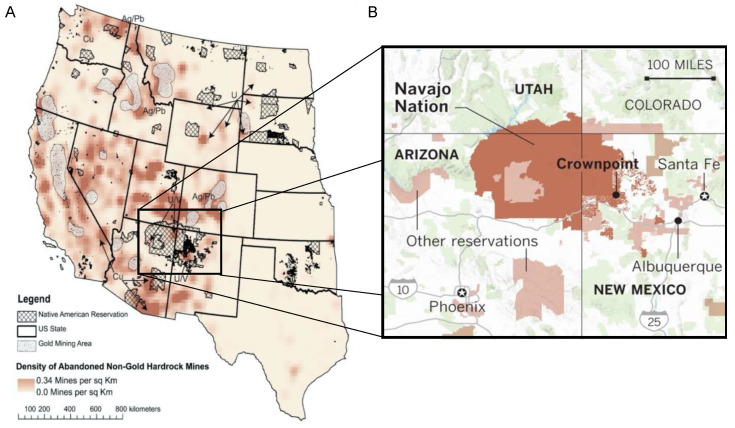
(**A**) Map of the western United States showing the locations of Native American reservations and the density of non-gold hard rock mines (adapted from ref [1]). (**B**) US Census Bureau map of Four Corners Region with Navajo Nation marked in dark brown: its area is approximately 27,000 square miles, roughly equal to that of West Virginia.

**Figure 2 ijerph-19-14939-f002:**
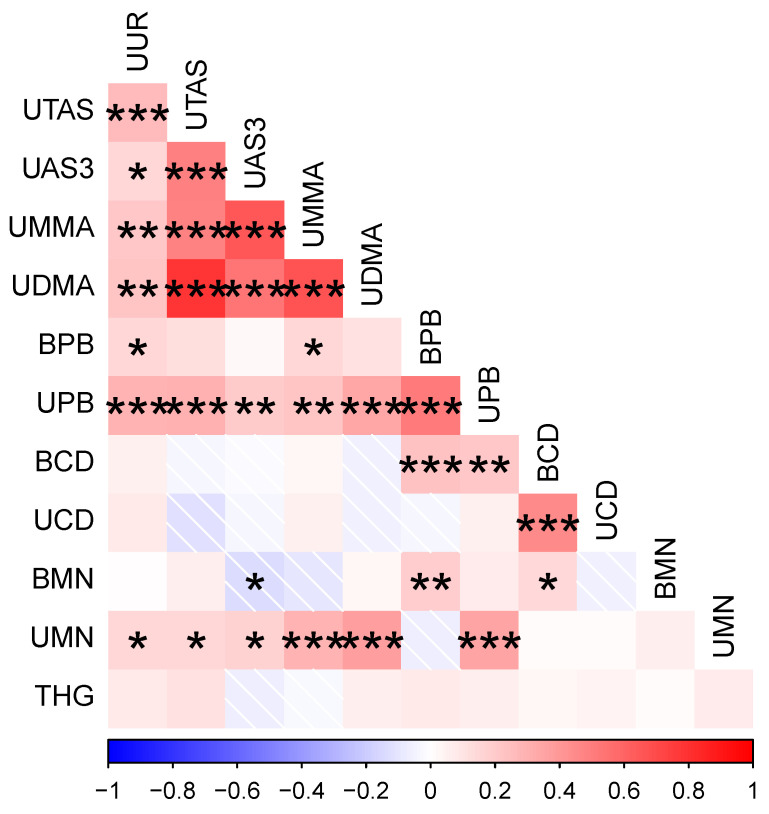
Spearman’s correlations between metal biomonitoring results. Correlation coefficient is designated by color with blues designating negative correlations and reds designating positive correlations. Asterisks denote significant correlations at the *p* < 0.05 (*), 0.01 (**), and 0.001 (***).

**Figure 3 ijerph-19-14939-f003:**
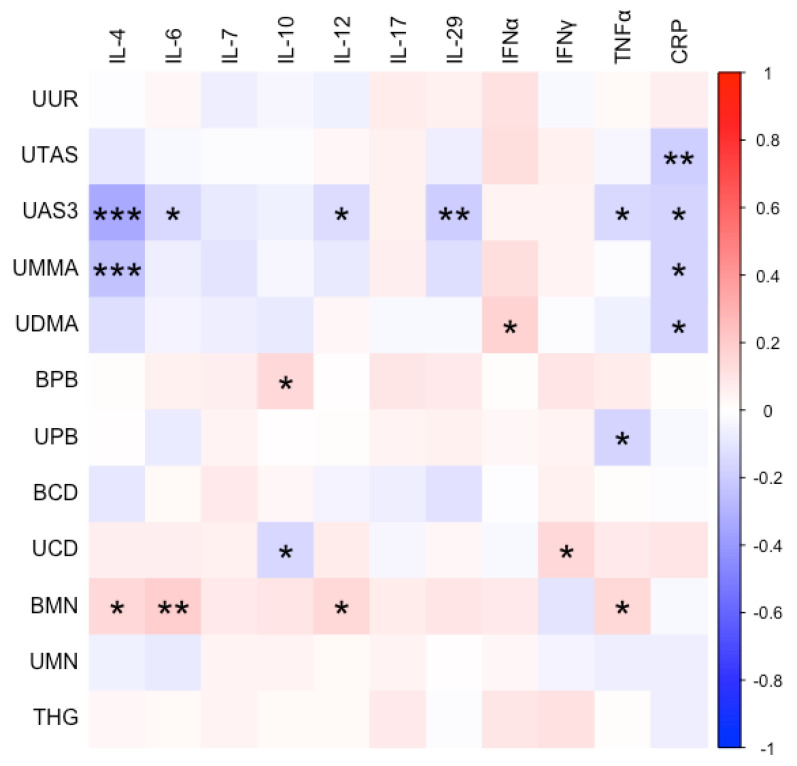
Spearman’s correlations between metal biomonitoring and biomarkers. Correlation coefficient is designated by color with blues designating negative correlations and reds designating positive correlations. Asterisks denote significant correlations at the *p* < 0.05 (*), 0.01 (**), and 0.001 (***).

**Table 1 ijerph-19-14939-t001:** Summary of sociodemographic characteristics of study cohort (*n* = 231).

Variable		
Maternal age (years)	Mean (SD):	27 (6.2)
	Category	Result: *n* (%) *
Education level	No high school diploma	97 (57.1)
	High school diploma	73 (42.9)
Household income	<$20,000/year	54 (39.4)
	>$20,000/year	83 (60.6)
Marital status	Married or living with a partner	112 (78.9)
	Not married or living with a partner	30 (21.1)
Pre-pregnancy BMI	Underweight or normal	52 (26.4)
	Overweight	70 (35.5)
	Obese	75 (38.1)
Trimester at sample collection	1st	34 (14.7)
	2nd	97 (42)
	3rd	100 (43.3)
Sex of child	Female	107 (49.3)
	Male	110 (50.7)

* Some data not available for all participants, % of participants with available data. NOTE: No difference in variables was observed between the subset of individuals selected for this study analysis and the NBCS as a whole.

**Table 2 ijerph-19-14939-t002:** Summary statistics for concentrations of exposure biomonitoring, including limit of detection (LoD).

Abbr.	Metal/Metabolite	Matrix	Units	*n*	LoD	Mean (SD)	Median (IQR)	NHANES Median (95th%Tile)	Factor Navajo vs. NHANES *
UUR	**Uranium**	Urine	µg/dL	216	0.002	0.03 (0.13)	0.02 (0.01–0.03)	0.005 (0.031)	**4×**
UTAS	Total arsenic	Urine	µg/dL	214	0.26	7.15 (6.12)	5.56 (4.15–8.08)	5.74 (49.9)	0.97
UAS3	**Arsenite (AsIII)**	Urine	µg/L	213	0.12	0.48 (0.39)	0.38 (0.19–0.62)	0.12 (1.11)	**3.17×**
UMMA	**Monomethylarsonic acid**	Urine	µg/L	213	0.2	0.52 (0.45)	0.38 (0.21–0.68)	0.28 (1.45)	**1.36×**
UDMA	**Dimethylarsinic acid**	Urine	µg/L	213	1.91	5.29 (3.61)	4.25 (3.07–6.52)	2.95 (12)	**1.44×**
BPB	Lead	Blood	µg/dL	212	0.07	0.41 (0.4)	0.32 (0.25–0.42)	0.88 (2.89)	0.36
UPB	Lead	Urine	µg/L	216	0.03	0.37 (0.56)	0.28 (0.19–0.39)	0.32 (1.38)	0.88
BCD	**Cadmium**	Blood	µg/L	212	0.1	0.34 (0.17)	0.31 (0.24–0.4)	0.27 (1.35)	**1.15×**
UCD	**Cadmium**	Urine	µg/L	216	0.036	0.25 (0.18)	0.22 (0.14–0.32)	0.179 (1.08)	**1.23×**
BMN	**Manganese**	Blood	µg/L	212	0.99	20.07 (6.82)	19 (15–24.64)	9.2 (16.1)	**2.07×**
UMN	**Manganese**	Urine	µg/L	215	0.13	0.37 (0.45)	0.23 (0.15–0.41)	0.13 (0.28)	**1.77×**
THG	Total mercury	Blood	µg/L	212	0.28	0.42 (0.27)	0.34 (0.2–0.51)	0.74 (4.66)	0.46

* Factor indicates the magnitude of Navajo medians relative to NHANES medians. Metals or metalloids in **bold** have a sample median value higher than the NHANES median.

**Table 3 ijerph-19-14939-t003:** Summary of significant (*p* < 0.05) Spearman’s correlations between cytokines and biomonitored metals and metalloids.

Cytokine	Metal	Rho	*p*	*n*
IL-4	UAS3	−0.34	0	211
	UMMA	−0.23	0.001	211
	BMN	0.15	0.033	209
IL-6	UAS3	−0.15	0.032	211
	BMN	0.18	0.009	209
IL-10	BPB	0.15	0.032	209
	UCD	−0.15	0.028	214
IL-12	UAS3	−0.14	0.047	211
	BMN	0.16	0.023	209
IL-29	UAS3	−0.19	0.006	203
IFNα	UDMA	0.18	0.011	203
IFNγ	UCD	0.14	0.035	214
TNFα	UAS3	−0.15	0.035	211
	UPB	−0.16	0.017	214
	BMN	0.16	0.023	209
CRP	UTAS	−0.18	0.009	204
	UAS3	−0.16	0.019	203
	UMMA	−0.17	0.018	203
	UDMA	−0.16	0.021	203

**Table 4 ijerph-19-14939-t004:** Summary of final multivariable models of the relationship between biomonitored metals and metalloids (UUR, UTAS, AsIII, DMA, and MMA, BPB, UPB, BMN, UMN, THG) and cytokines after variable selection.

Cytokine	*n*	Predictors Retained in Multivariate Model	Estimate	Std Error	*p*
IL-4	203	age	0.013	0.008	0.114
		**log(UAS3)** *	**−0.228**	0.058	0
		**log(BCD)**	**−0.261**	0.113	0.022
		**log(BMN)**	**0.31**	0.156	0.049
IL-6	203	log(UAS3)	−0.125	0.065	0.054
		log(BMN)	0.329	0.172	0.057
IL-7	203	log(BCD)	0.1	0.061	0.103
IL-10	203	age	−0.02	0.008	0.017
		log(UDMA)	−0.183	0.098	0.064
		log(UPB)	−0.176	0.12	0.143
		**log(BPB)**	**0.422**	0.134	0.002
		log(UMN)	0.139	0.075	0.066
IL-12	203	log(UTAS)	0.17	0.107	0.112
		log(UAS3)	−0.113	0.069	0.103
		log(BCD)	−0.248	0.132	0.062
		log(UCD)	0.142	0.088	0.107
		**log(BMN)**	**0.386**	0.17	0.024
IL-17	203	**log(BPB)**	**0.3**	0.128	0.02
		log(BCD)	−0.266	0.144	0.066
IL-29	196	**log(UAS3)**	**−0.552**	0.181	0.003
		**log(UPB)**	**0.545**	0.258	0.036
		**log(BCD)**	**−0.996**	0.392	0.012
		**log(UCD)**	**0.505**	0.253	0.047
		log(BMN)	0.778	0.478	0.105
IFNα	196	log(UTAS)	−0.51	0.305	0.096
		**log(UDMA)**	**0.979**	0.316	0.002
		log(UMN)	−0.221	0.157	0.16
		log(THG)	0.284	0.199	0.155
IFNγ	203	trim.L	−0.248	0.112	0.028
		trim.Q	−0.05	0.094	0.595
		log(UUR)	−0.096	0.062	0.123
		**log(BPB)**	**0.255**	0.105	0.017
		log(THG)	0.173	0.094	0.066
TNFα	203	age	−0.007	0.005	0.164
		**log(UPB)**	**−0.183**	0.06	0.003
		**log(BPB)**	**0.176**	0.074	0.017
		**log(BCD)**	**−0.147**	0.071	0.04
		**log(UCD)**	**0.152**	0.053	0.004
		**log(BMN)**	**0.186**	0.089	0.037
CRP	171	**log(UTAS)**	**−0.809**	0.24	0.001
		**log(UMMA)**	**−0.324**	0.162	0.047
		**log(UDMA)**	**0.667**	0.289	0.022
		log(BPB)	0.273	0.171	0.111

* Strong immunomodulatory effects (*p* of estimate < 0.05) are in bold.

## Data Availability

In accordance with Tribal Sovereignty, any data connected with biological samples from tribal members can only be shared with appropriate tribal policies and permissions. Should other researchers wish to use these data at a future time for reanalysis or to explore additional hypotheses, they would need to request access to the data through application to the Navajo Nation Human Research Review Board, and their proposal would need to meet the guidelines for research in accordance with tribal policy.

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
