# Peer review of "Chronic Community Exposure to Environmental Metal Mixtures Is Associated with Selected Cytokines in the Navajo Birth Cohort Study (NBCS)"

_ijerph, 2022, doi:10.3390/ijerph192214939_

Round 1

Reviewer 1 Report

The manuscript deals with the immune responsiveness of several metals through environmental exposure on Navajo Nation community members. The topic is in the field of environmental immunology or immunotoxicology, being accepted that exposure to environmental contaminants can produce profound effects on the immune system, leading to inadequate or excessive production of inflammatory cytokines.

The manuscript is robust, using a large battery of tests. The topic is interesting, the experiments seem properly performed and the results correctly interpreted.

The followings should improve the quality of the manuscript:

Please indicate in the text the producer of the instrumentation used in metals analysis If a HPLC was used in combination with ICP-MS for the analysis of arsenic compounds, please mention it. Also, the chromatographic column used for separation.

The authors should detail the QA/QC of metal analysis. If certified reference materials were used, please add them in the text. 

Author Response

The manuscript deals with the immune responsiveness of several metals through environmental exposure on Navajo Nation community members. The topic is in the field of environmental immunology or immunotoxicology, being accepted that exposure to environmental contaminants can produce profound effects on the immune system, leading to inadequate or excessive production of inflammatory cytokines.

The manuscript is robust, using a large battery of tests. The topic is interesting, the experiments seem properly performed and the results correctly interpreted.

Authors: Thank you for your helpful feedback and comments. 

The followings should improve the quality of the manuscript:

Please indicate in the text the producer of the instrumentation used in metals analysis.

Authors: Added in revised lines 140-143.

If a HPLC was used in combination with ICP-MS for the analysis of arsenic compounds, please mention it. Also, the chromatographic column used for separation.

Authors: Added in revised lines 141-143.

The authors should detail the QA/QC of metal analysis. If certified reference materials were used, please add them in the text. 

Authors: Added in revised lines 147-151.

Reviewer 2 Report

Dear Editor and Authors

The manuscript is well written and presents a very interesting study. The biggest concern is that part of the results are based on the determination of trace elements, but there is no information on analytical quality control. Therefore, I request a major review before publication in IJERPH.

Arsenium is not considered a metal, I suggest using the term trace elements when the authors refer to exposure to all elements.

Figure 1 must be presented horizontally.

Metal biomonitoring: authors must present limits of quantification (LOQ) of elements. Analytical methodologies to determine the elements should be presented briefly. What certified reference materials were used? The recovery percentage must be displayed.

Author Response

Dear Editor and Authors

The manuscript is well written and presents a very interesting study. The biggest concern is that part of the results are based on the determination of trace elements, but there is no information on analytical quality control. Therefore, I request a major review before publication in IJERPH.

Authors: We thank the reviewer for their helpful feedback to improve the paper.

Arsenium is not considered a metal, I suggest using the term trace elements when the authors refer to exposure to all elements.

Authors: We have clarified in the first line of the manuscript that we use the term “metals” to refer to metals and metalloids collectively, however we add “metalloids” in specific areas throughout the article where arsenic and its species are highlighted and in the conclusions.

Figure 1 must be presented horizontally.

Authors: Changed.

Metal biomonitoring: authors must present limits of quantification (LOQ) of elements.

Authors: The range of limits of detection of elements is listed in line 152. In addition, we now provide limits of detection by element in Table 2.

Analytical methodologies to determine the elements should be presented briefly. 

Authors: Our metal biomonitoring analysis was conducted by the CDC according to their standard and published protocols, which describe methodologies in detail and which we cite as references 50-55. For greater transparency, however, we now further explain HPLC analysis for arsenic speciation, provide details on equipment and manufacturers, briefly explain QA/QC procedures, and explicitly list CDC protocol numbers parenthetically. These edits are throughout the methods section “Metal Biomonitoring”.

What certified reference materials were used?

Authors: We have added quality control methods for the metal analysis in revised lines 147-151.

The recovery percentage must be displayed.

Authors: We add information on CDC acceptable range of percentage recovery in revised lines 153-154.

Round 2

Reviewer 2 Report

No comments.